# The Patient’s Point of View: COVID-19 and Neuroendocrine Tumor Disease

**DOI:** 10.3390/cancers14030613

**Published:** 2022-01-26

**Authors:** Sebastian Krug, Maryam Khosravian, Julia Weissbach, Katharina George, Marko Damm, Jakob Garbe, Jens Walldorf, Philipp A. Reuken, Tania Amin, Alexander Siebenhüner, Jonas Rosendahl, Thomas M. Gress, Patrick Michl, Jörg Schrader, Anja Rinke

**Affiliations:** 1Clinic for Internal Medicine I, Martin-Luther University Halle/Wittenberg, Ernst-Grube-Straße 40, D 06120 Halle, Germany; sebastian.krug@uk-halle.de (S.K.); maryam.khosravian@uk-halle.de (M.K.); julia.weissbach@uk-halle.de (J.W.); katharina.george@uk-halle.de (K.G.); marko.damm@uk-halle.de (M.D.); jakob.garbe@uk-halle.de (J.G.); jens.walldorf@uk-halle.de (J.W.); jonas.rosendahl@uk-halle.de (J.R.); 2Department of Internal Medicine IV, Jena University Hospital, D 07747 Jena, Germany; philipp.reuken@med.uni-jena.de; 3I. Medical Department, Uniersity Medical Center Hamburg-Eppendorf, Martinistrasse 52, D 20246 Hamburg, Germany; t.amin@uke.de (T.A.); jschrader@uke.de (J.S.); 4Department of Medical Oncology and Hematology, University Hospital Zurich and University of Zurich, Rämistrasse 100, CH-8091 Zurich, Switzerland; Alexander.Siebenhuener@usz.ch; 5Department of Gastroenterology and Endocrinology, University Hospital Marburg, Baldinger Strasse, D 35043 Marburg, Germany; gress@med.uni-marburg.de (T.M.G.); sprengea@uni-marburg.de (A.R.)

**Keywords:** NET, neuroendocrine, COVID-19, SARS-CoV-2, survey, vaccination, anxiety, worries, psychological factors

## Abstract

**Simple Summary:**

Since neuroendocrine tumor patients require a highly specialized and interdisciplinary infrastructure for diagnostic and therapy, medical care has been very challenging during the COVID-19 pandemic. In cooperation with the patient organization NETZWERK NeT we were able to distribute a comprehensive survey, which has profoundly investigated the healthcare structure and patient-specific concerns during the crisis. In addition to regular medical care, there is a considerable need to measure patient-reported outcomes such as social and emotional distress in a structured way to optimize individual therapy for NET patients.

**Abstract:**

The assessment of cancer patient care during the COVID-19 pandemic has been mainly reported from a physician’s perspective. Patients with rare tumor entities such as neuroendocrine tumors (NET), which require a complex and specialized care infrastructure, were highly affected by the COVID-19 crisis. Using a structured questionnaire consisting of a general section on the disease and a special COVID-19 section to record medical care, vaccination behavior as well as social and psycho-emotional parameters were collected from NET patients. The survey was distributed via direct medical contact and via the patient organization NETZWERK NeT. A total of 684 patients participated in the survey and 79.2% (*n* = 542) of the participants answered the questionnaire completely (54 questions). Patient characteristics were comparable to those in large NET registries. The majority of participants were patients with pancreatic and small bowel NET on somatostatin analogue (SSA) therapy. Medical care under COVID-19 was adequate and appointment cancellations and postponements were not common. Nevertheless, the majority of patients were worried about adequate treatment for their tumor disease during the crisis. Most of the participants considered themselves to be at risk of severe COVID-19 infection and were therefore very concerned. This was accompanied by an extremely high vaccination readiness rate of 90%. Increased distress in the social and psycho-emotional domains in the course of the crisis reflected a need for optimization in the medical care of NET patients, although the rate of COVID-19 positive participants was low (3.7%). Therefore, patient-reported measurements are required to identify and address all areas of medical care. Overall, our survey provides an essential contribution to the care of NET patients during the COVID-19 pandemic from the patient’s perspective.

## 1. Introduction

The Severe Acute Respiratory Syndrome Coronavirus 2 (SARS-CoV-2—COVID-19) has been a global health emergency since January 2020. Although vaccination is now widely available, infection numbers are rising again. Apart from the severe economic consequences of the pandemic, various aspects of human coexistence, traditions and habits have changed dramatically. Due to the pandemic, global health care systems have been challenged as never before [1,2]. The number of COVID-19 positive cases is still increasing and currently surpasses 200 million worldwide, with the highest numbers in America and Europe. Over four million deaths have been directly related to COVID-19, with the number of unreported cases being significantly higher (WHO dashboard).

Patients with cancer were particularly affected during the pandemic. There are many different explanations for this phenomenon, including worse survival rates of cancer patients suffering from COVID-19 infections, deterioration of medical care through prioritization and resource limitations for both in- and outpatients and a delay in initial cancer diagnosis due to COVID-19 fear [3,4,5].

The European Society of Medical Oncology (ESMO) offered recommendations for cancer care in general and guidance for diagnostics and therapy in the COVID-19 era for some tumor entities. Independently, practical recommendations for patients with neuroendocrine neoplasms (NEN) were provided [6]. In these rare neoplasms with a very heterogeneous spectrum ranging from very slow-growing neuroendocrine tumors to aggressive neuroendocrine carcinomas, highly individualized and complex medical care is required. Factors to be considered include functionality (clinically relevant hormone secretion), primary tumor localization, Ki-67 proliferation index and somatostatin receptor status. Highly specialized centers, including centers of excellence with multidisciplinary teams and tumor boards (MDT), are urgently required for the optimal treatment of this disease. In the context of the pandemic, we have observed strong regional differences. The Italian Association for Neuroendocrine Tumors (It.a.net) indicated a significant influence of the COVID-19 pandemic on the number of newly diagnosed NEN patients, surgical procedures for these patients, nuclear medicine therapies (PRRT) and MDT activities [7]. In contrast, data from Germany, Austria, and Switzerland demonstrated only a minority of postponed or canceled diagnostic and therapeutic procedures. In addition, there was a significant difference in the assessment of how COVID-19 impacted patient care between physicians in a university and a non-university setting [8].

However, all assessments were presented from a physician’s perspective. A structured survey from the patient’s perspective has not been reported yet. In collaboration with the patient organization NETZWERK NeT (Netzwerk Neuroendokrine Tumoren (NeT) e.V.; https://www.netzwerk-net.de, accessed on 30 November 2021), we conducted in a structured manner a comprehensive analysis of the management of NEN patients during the COVID-19 pandemic, including psychological parameters such as worries and fears of these patients as well as their vaccination behavior.

## 2. Methods

Ethical approval was obtained from the local ethical review committee of the Martin Luther University Halle-Wittenberg (number: 2021-015; January 2021). A total of 54 questions was developed, including 19 general questions about characteristics of the underlying NET disease, disease history, actual treatment status, symptoms and 35 specific questions about the impact of the COVID-19 pandemic on tumor treatment and disease care. Multiple choice or yes/no answers and optional text answers were available in the survey. Except for the optional text boxes, all questions were mandatory. Participants were able to temporarily save the questionnaire before completion and complete it at a later time.

The survey was performed between March 19th and June 19th 2021 in the following three German-speaking countries: Austria, Switzerland and Germany (Appendix A). The survey was open to patients with NET disease undergoing therapy (including watch and wait) or follow-up. The survey was circulated via the patient organization NETZwerk NET e.V. as well as via personal contact with the attending NET specialist. The full survey is provided in the Appendix A.

LimeSurvey software (LimeSurvey GmbH, Hamburg, Germany) was used to conduct the online survey. Descriptive statistics were calculated using Office Excel 2016 (Microsoft Corporation, Redmond, DC, USA) and GraphPad Prism 5 (GraphPad Software, San Diego, CA, USA). The association between two variables was performed based on Chi-square and Fisher’s exact tests, as appropriate.

## 3. Results

### 3.1. Patient and Tumor Characteristics

A total of 684 patients participated in the survey. Of them, 79.2% (*n* = 542) answered all questions completely (see Table 1). Comprehensive data analysis was performed only on participants with complete data sets. There were 523 patients (96.4%) from Germany, 10 from Austria (1.9%) and 9 from Switzerland (1.7%). The largest group of patients were between 61–80 years of age (*n* = 257, 47.4%), followed by the groups between 41–60 (*n* = 254, 46.9%), 18–40 (*n* = 22, 4.1%), and >80 years of age (*n* = 9, 1.7%). Three-quarters of the patients were married (*n* = 412, 76.0%) and one-quarter were either divorced, widowed, or single (altogether *n* = 125, 23.1%). The distribution of primary tumor localization was as follows: small bowel (jejunum, ileum) 39.1% (*n* = 212), pancreas 24.7% (*n* = 134), duodenum 9.2% (*n* = 50), unknown primary site 7.8% (*n* = 42), lung 7.6% (*n* = 41), colorectal 4.8% (*n* = 26) and stomach 2.8% (*n* = 15). In 25 patients, rare localizations were present or specific information could not be obtained (4.6%). A total of 217 cases (40.0%) were functionally active (FA-NET), 182 cases (33.6%) were functionally inactive and in 143 cases (26.4%) functionality was unknown. Among FA-NET cases, most patients presented with carcinoid syndrome (76.9%, *n* = 167), and considerably fewer with gastrinoma (6.5%, *n* = 14), insulinoma (2.8%, *n* = 6) or rare entities such as glucagonoma (1.8%, *n* = 4) or somatostatinoma (0.9%, *n* = 2). The time of diagnosis was >5 years ago in 250 participants (46.1%). In 40.8% (*n* = 221) of the cases the disease was diagnosed within the last 1–5 years or within the last 12 months in 13.1% (*n* = 71). The time interval between onset of symptoms and diagnosis varied widely, ranging from <3 months in 23.4% (*n* = 127), 3–6 months in 13.3% (*n* = 72), 7–12 months in 9.4% (*n* = 51), 13–24 months in 14.4% (*n* = 78) and >24 months in 21.6% (*n* = 117). A total of 97 participants were unable to provide any information. Two-thirds of patients had ongoing symptoms at the time of survey completion (68.8%, *n* = 373), including impaired resilience (67.0%, *n* = 250), weight loss (24.7%, *n* = 92), nausea/vomiting (15.0%, *n* = 56), diarrhea (59.3%, *n* = 221), pain (33.8%, *n* = 126), loss of appetite (8.9%, *n* = 33), mental discomfort (39.9%, *n* = 149) and flush (29.2%, *n* = 109).

### 3.2. Treatment

Most patients had been treated in university centers and ENETS centers (71.0%, *n* = 385), followed by specialized physicians (17.2%, *n* = 93) and non-university hospitals (11.8%, *n* = 64). When asked about their present disease status, 68.9% (*n* = 369) of the participants reported to have active tumor disease (current treatment or watch-and-wait), while the remaining patients were undergoing follow-up care (see Table 1). The therapy most frequently used was somatostatin analogues (SSA) (54.5%, *n* = 201), watch-and-wait (18.4%, *n* = 68), peptide receptor radionuclide therapy (PRRT) (7.9%, *n* = 29), chemotherapy (7.9%, *n* = 29) and everolimus (4.1%, *n* = 15). Therapy-specific adverse events were rarely reported and were specified as moderate to severe in only 24.4% (*n* = 132) and 6.5% (*n* = 35), respectively. Most participants were rather satisfied (36.4%, *n* = 197) and very satisfied (51.5%, *n* = 279) with the medical treatment, the appointment coordination and the medical care. About 10% (*n* = 55) expressed some dissatisfaction or no satisfaction at all. Actually, 23.8% (*n* = 129) of the participants were unable to work due to their NET disease.

### 3.3. Management of NET Patients during COVID-19

Of the 542 participants, 151 reported (27.9%) that the COVID-19 pandemic had affected their therapy and controls (Figure 1). Outpatient appointments were postponed or cancelled more often than inpatient appointments (12.4%, *n* = 67 vs. 6.6%, *n* = 36). The following interventions were referred to: MRI/CT (9.0%, *n* = 49), PET/CT (6.5%, *n* = 35), surgical procedures (5.5%, *n* = 30), supportive therapies (4.1%, *n* = 22), PRRT (1.3%, *n* = 7) and chemotherapy (0.9%, *n* = 5). Although absolute numbers for postponed/canceled treatments with PRRT and chemotherapy were low, up to 20% of patients on active treatment with PRRT and chemotherapy were affected by changes in the treatment schedule. Moreover, outpatient and inpatient appointments were not planned in 16.4% and 53.3% of cases during this period, respectively. Telephone or video consultations were provided in 32.8% of the cases (*n* = 178). The personal accessibility of the attending NET specialist during the COVID-19 pandemic remained unchanged for 75% (*n* = 407) of the participants. However, in 11.3% (*n* = 61) and 4.2% (*n* = 23) of cases, the physician was less and considerably less accessible, respectively.

### 3.4. Anxieties and Concerns of Patients during COVID-19

Regardless of whether receiving systemic therapy or being on follow-up, 78.6% (*n* = 426) of participants considered themselves to be at risk of a severe COVID-19 infection. Of the participants, 26.9% (*n* = 146) discussed their personal risk of severe COVID-19 infection with their NET specialist. In this context, 16.2% (*n* = 88) of the respondents (60.3% of 146 patients) mentioned that they had been informed about interventions to reduce the risk of exposure to COVID-19. The vast majority of participants (86.7%, *n* = 470) reported wearing FFP2 masks for self-protection; a surgical mask was used by only 9.8% (*n* = 53) of participants. In addition, many patients applied self-testing once or several times for COVID-19 (60.3%, *n* = 382). Some participants (4.2%, *n* = 23) reported that they would have preferred a COVID-19 test. Reasons for the lack of implementation were not recorded.

Of the participants, 45.9% (*n* = 249) and 36.3% (*n* = 197) were hesitant to visit the NET centers for outpatient or inpatient treatment as a result of the COVID-19 situation (Figure 2). In subgroup analyses, baseline characteristics such as tumor-specific symptoms, age, hormone secretion and therapy status were assessed as factors relevant for the avoidance of health-care facilities. Among outpatients (baseline fear 45.9%), symptoms (48.8%), age over 60 (46.6%) and recent follow-up (47.9%) tended to be associated with greater fear of COVID-19 infection (Figure 3A). For the inpatients (baseline fear 36.3%), this was the case for symptoms (40.2%), functional activity (37.8%) and therapy (37.7%) (Figure 3B). Outpatient or inpatient follow-up consultations were canceled by patients themselves in 10.5% (*n* = 57) and 1.9% (*n* = 10) of cases, respectively, due to fear of COVID-19 infection.

Overall, 34.9% (*n* = 162) of the participants feared worsening of their NET disease during the COVID-19 pandemic. In 1.3% (*n* = 7) of the patients, a severe/very severe deterioration had already occurred, while a minor deterioration was reported in 13.1% (*n* = 71) of the patients (Figure 4).

Regarding the handling of private and/or family meetings by the patients during the pandemic, 14.9% (*n* = 81) answered that all meetings and 47.2% (*n* = 256) that many meetings had been canceled. Only 9.9% (*n* = 54) of the participants reported that no appointments had been canceled. Of the participants, 61.3% (*n* = 333) had regularly attended in-person meetings of NET patient organizations before the COVID-19 pandemic had started. Of these, 47.4% (*n* = 257) reported that these meetings no longer were held in person (Figure 1).

### 3.5. Psychological Impact of the COVID-19 Pandemic for NET Patients

As already mentioned, 89.7% (*n* = 485) of the participants expressed anxieties related to the pandemic, 54.2% (*n* = 294) had minor anxieties, 35.2% (*n* = 191) had major anxieties) and 78.6% (*n* = 426) of the participants regarded themselves as a risk group for severe COVID-19 disease.

To better stratify the psychological impact of COVID-19 pandemic upon the participants, we formed three groups of participants: patients under therapy (55.6%, *n* = 301), patients under a watch-and-wait strategy (12.5%, *n* = 68) and patients under surveillance (31.9%, *n* = 179) (Figure 5A). Interestingly, the subgroup of patients under surveillance without active treatment showed the highest worries concerning COVID-19 (90.2%), followed by the therapy patients (90.1%) and the watch-and-wait group (85.3%) (Figure 5B). Consequently, the surveillance group also demonstrated the highest attributed risk for severe COVID-19 infection (82.1% compared to 77.1% (therapy group) and 76.5% (watch-and-wait group)) (Figure 5C).

Overall, 58.1% (*n* = 315) of the respondents felt more socially isolated due to the COVID-19 situation (from least to most affected). As before, the surveillance group achieved the highest values in the subgroup analyses (53.2%) (Figure 5D). Strategies to overcome social isolation were as follows: support from family (49.6%, *n* = 269), support from friends (13.5%, *n* = 73) and support from neighbors (6.1%, *n* = 33).

Psychiatric comorbidities such as depression, anxiety disorders or unstable emotional personality disorders were reported by 20.7% of all participants (*n* = 112). Within the distinct treatment groups, there was a significant difference between the surveillance group and the therapy plus watch-and-wait cohort (30.1% vs. 11.9% and 13.2%) (Figure 5E). In the context of the COVID-19 crisis, these comorbidities worsened severely or very severely in 11.6% (*n* = 63) of cases. Forty-seven participants reported no or only minor negative effects.

In a panel of five questions, we asked about a number of items associated with emotional state. The answer options for these questions ranged from: yes, completely or rather agree; partly agree; no, rather disagree; or not at all. Lack of happiness, loss of interest, depressed mood and reduced motivation were observed in 56.6% (*n* = 307) of all participants. In relation to the different treatment groups, there was no difference concerning this item (Figure 5F). Furthermore, insomnia in 54.8% (*n* = 297), loneliness in 55.2% (*n* = 299), alterations of the body feeling/sensation in 46.5% (*n* = 252) and loss of hobbies in 76.8% (*n* = 416) of participants were frequently recorded for the entire study population.

Additionally, there were significant correlations between worsening of the disease or fear of deterioration and signs of a depressive episode or existing psychiatric disorders (Fisher’s exact test: 250/292 vs. 112/430; *p* < 0.001 and 250/292 vs. 307/235; *p* < 0.001).

### 3.6. COVID-19 Positive Patients and Their Characteristics

Within the survey, 20 participants (3.7%) reported that they experienced a COVID-19 infection. We examined this subgroup in detail and stratified it in patients in active therapy (*n* = 11) or patients under follow-up (*n* = 9) (Table 2). Half of the patients with COVID-19 infection had a pancreatic primary tumor, which was treated in 11 patients at time of COVID-19 diagnosis as follows: SSA in 5 patients, PRRT in 2 patients, TKI in 1 patient, chemotherapy in 1 patient, and watch-and-wait strategy in 2 patients. The most common comorbidities included hypertension, diabetes, lung disease, chronic renal failure and liver cirrhosis (see Table 2). In two patients no relevant comorbidities were mentioned. Interestingly, preexisting lung disease was exclusively reported in four patients under active systemic treatment. A total of 15 of the 20 patients had classified themselves as being at risk for severe COVID-19 infection. Based on self-assessment, 9 infections were mild, 10 were moderate and 1 was severe, requiring intensive care. The latter was observed under PRRT in a patient with small bowel NET. Subsequent symptoms of the COVID-19 infection were reported by those affected as follows: fatigue (45%, *n* = 9), impaired concentration (35%, *n* = 7), impaired olfactory/gustatory sense (20%, *n* = 4), insomnia (10%, *n* = 2), and headaches (10%, *n* = 2). A total of 5 participants (25%) reported no symptoms. At the time of the survey, 17 participants in the COVID-recovered group were planning to get subsequently vaccinated against COVID-19 and 3 were still undecided.

### 3.7. Opinions on Vaccination

In addition to the 20 patients who had recovered from COVID-19 at time of the survey, a total of 29.5% (*n* = 160) of the participants reported COVID-19 positive individuals in their close environment. The willingness to get vaccinated was 89.3% (*n* = 484) at the time of the survey (72.4% of the participants received an influenza vaccination within the last 5 years). We compared our data (published by our group at the beginning of the year 2021 [9]) with data obtained by similar online surveys on healthy control persons (*n* = 410) and patients with inflammatory bowel disease (IBD; *n* = 1032). Interestingly, the willingness to get vaccinated for either influenza or COVID-19 was significantly lower in these other two patient cohorts (controls: 45.3% and 55.6%; IBD patients: 65.1% and 58.5%) (Figure 6). Of the participants in our NET cohort, 7.0% (*n* = 38) were still undecided and 3.7% (*n* = 20) refused vaccination. Notably, the vaccination refusal rate in the NET cohort was significantly lower compared to the independently reported cohorts of healthy control subjects (13.2%, *n* = 54) and IBD patients (11.1%, *n* = 114). Fourteen NET patients (2.6%) were in doubt about COVID-vaccination due to a lack of long-term data and insufficient safety aspects. Two patients were concerned that the vaccination would interfere with current therapy and might worsen therapy efficacy. One patient was discouraged from getting vaccinated by a physician’s advice due to poor general condition and another patient described a known allergy to vaccination.

## 4. Discussion

Taking care of patients with neuroendocrine neoplasms represents a major challenge in the COVID-19 era. In this respect, studies have observed considerable regional differences in the care of patients, both in the outpatient and inpatient sectors [6,7,8]. However, all published empirical data so far have been based exclusively on physicians’ assessments. This study investigated the impact of the COVID-19 pandemic from the patient’s perspective in a structured setting. Our data show that many NET patients considered themselves at risk for a severe course of COVID-19 infection and were therefore very concerned. Medical accessibility during and education about the pandemic by the attending physician was usually provided. However, NET patients displayed a high level of psychological stress and anxiety, which had not been adequately addressed. Compared to the general population, there was a very high willingness to get vaccinated, despite the fact that the risk for COVID-19 infection is not higher in this group of patients than in other chronic diseases.

The participants of this survey can be considered as representative of NEN patients in general in central Europe. Although this survey addressed a specific population of NEN patients organized through a patient organization and treated mainly in university hospitals and ENETS centers, it displays a similar age distribution and primary tumor localization compared to the German NET registry [10]. Compared to these registry data, a slightly higher proportion of patients with small intestine (duodenal and ileum) NEN participated in our survey and, additionally, patients with lung NEN were included. However, compared to the data of Niederle et al., there were significantly fewer participants with stomach, appendix and rectum NEN [11]. Due to the mostly favorable long-term prognosis of localized stages of these entities, these patients are usually followed-up in a less structured way [1]. Interestingly, about half of the participants reported having a functional disease, mainly the carcinoid syndrome (CS). Within all NENs, the functionality is significantly less frequent than reported in this survey [12]. The higher rate of functional tumors observed in our cohort is in line with the more favorable short diagnostic latency between symptom onset and diagnosis reported in our cohort compared to international surveys, in which estimated median latencies of more than 50 months were observed [13,14].

NEN patients require a multidisciplinary team with an appropriate armamentarium of diagnostic and therapeutic approaches. Therefore, most patients are treated in university hospitals or dedicated NET centers. During the COVID-19 pandemic, these specialized hospitals were and still are highly affected by governmental regulations prioritizing COVID-19 positive patients. Interestingly, patients reported that only a minority of outpatient or inpatient appointments had to be postponed [8]. One reason for the continuity of medical care was certainly the implementation of telephone or video consultations, which was used by one-third of the participants during the COVID-19 period. This approach is supported by the current ESMO guidelines aiming to avoid personal visits [15]. Furthermore, patients who required only follow-up investigations or patients who received somatostatin analogues or watch-and-wait strategies within our survey were more flexible and did not rely on frequent control visits or inpatient treatment at centers. Still, a substantial proportion of patients on active treatment requiring regular inpatient and outpatient contacts (e.g., PRRT, chemotherapy) reported changes in the treatment schedule. This might have contributed to the fear of disease worsening during the COVID pandemic.

The survey included 20 patients with SARS-CoV-2 infection, one of whom required intensive care. Thus, the overall rate of COVID-19 positive participants was 3.7%. This is significantly higher than the infection rate of 0.68% evaluated in patients with active cancer therapy in Italy [16]. However, our survey covered the period from the start of the COVID-19 pandemic in Europe to June 2021, and thus recorded a long observational phase, which may explain the higher rate of COVID-19 infections. When compared to the infection rate of the entire population of Germany at this time (3,728,141 cases on 1 June 2021) of approximately 4.5%, there is no apparent difference [5]. In our view, the impact of severe and lethal COVID-19 infections is not well reflected by this survey. However, based on the results of the INTENSIVE study, which collected COVID-19 positive GEP-NEN patients worldwide, it is estimated that this ratio is low in GEP-NEN patients [17]. In the INTENSIVE study, 89 COVID-19 positive NEN patients were collected worldwide. Overall, only few severe courses of the infection were observed: 7 patients (7.8%) died due to COVID-19, while 80% of the patients completely recovered without long-term side effects [17]. The latter is not reflected in our study by the COVID-19 recovered participants. The vast majority of participants, 75%, reported long-term impairments such as fatigue, impaired concentration and senses, insomnia and headaches. It is apparent that patient-reported outcomes provide a different, possibly more accurate reflection in comparison to medical records. Altogether, our data indicate that NEN patients have no increased susceptibility for SARS-CoV-2 infections or a severe course of the disease compared to the general population.

Early vaccination is recommended because patients with regular out- and inpatient appointments, active cancer treatment and frequent contacts with health care workers are at risk for COVID-19 [18]. Only little information is available on the acceptance and willingness to get vaccinated in the entire population and particularly among cancer patients. In an analysis performed in the United States, 69% of adult participants wanted to be vaccinated [19]. Similar results were observed in Germany, where 65.1% of healthy adults planned to get vaccinated as soon as possible [9]. A recently published nationwide multicenter survey in Korea resulted in a vaccination readiness of 61.8% in cancer patients, higher than other reports from France and Poland (53.7 and 60.3%), but lower compared to the general population data [2,20,21]. In our study, the acceptance and willingness to vaccinate was 89.3% and thus much higher compared to other cancer patient cohorts. Possible reasons include a generally increased willingness to get vaccinated (see also influenza vaccination readiness), adequate medical education and the self-assessment to be at risk for a severe COVID-19 infection and solicitude about the pandemic. Most likely, the most influential factor to convince undecided patients is the physician’s advice [20].

The measurement of health-related quality of life (HRQoL) in patients with cancer is essential for the correct assessment of symptoms related to tumor burden and therapy. In NEN, further characteristics have to be taken into account: hormone secretion and its concomitant symptoms, long/chronic course of the disease, possibly a hereditary trait with familial predisposition, as well as a broad spectrum of therapies with various side effects. Therefore, the assessment of HRQoL is particularly challenging. Previous work has demonstrated that depression and anxiety are very relevant in NEN patients regardless of hormone secretion [22,23]. In our study, a great fear of acquiring a COVID-19 infection and the worsening of NEN disease in the course of the COVID-19 pandemic were detected. Given the reduced availability of contacts to the NET center and the treating physician, these worries might not have been adequately addressed. Compared to a study prior to the COVID-19 pandemic, which reported a 25% rate of anxiety in advanced NEN stages, we observed a significant increase in the COVID-19 pandemic [23]. In parallel, major symptoms of depression were reported in 50–75% of patients in our survey. Since most participants were on surveillance or SSA treatment, the therapy modality is most likely not a major influencing factor. Rather, we believe that the self-assessment of being at increased risk resulting in anxiety and subsequent psycho-emotional distress is a specific pattern for NEN patients during the course of the COVID-19 crisis. Interestingly, the prevalence of anxiety and depression also seems to increase over the course of the pandemic waves. In an Italian study by Lauricella, this effect was shown within the 1st and 2nd pandemic waves for NET patients [24]. Since our survey was conducted during the 3rd wave, the higher rates of psycho-emotional distress can be better explained.

In addition to disease-focused medical care, the structured use of psycho-oncological support such as video or telephone consultations can help alleviate patients’ concerns. Patient-reported outcome measurements of physical, social, and emotional distress should therefore be regularly recorded, evaluated in a structured way and addressed in order to be able to offer NEN patients a comprehensive and individualized therapy.

## 5. Conclusions

The most important finding of our survey is the increased anxiety reported by NEN patients. This includes fear of the disease worsening as well as fear of acquiring a severe COVID-19 infection. This fear may explain the high rate of willingness to get vaccinated. These highly relevant patient concerns are frequently not adequately addressed by physicians. Thus, we conclude that actively addressing the patients’ anxieties during the consultation is eminently important. Providing adequate, individualized information about the risk of severe COVID-19 disease and ensuring access to optimal cancer care will greatly benefit patients’ well-being.

## Figures and Tables

**Figure 1 cancers-14-00613-f001:**
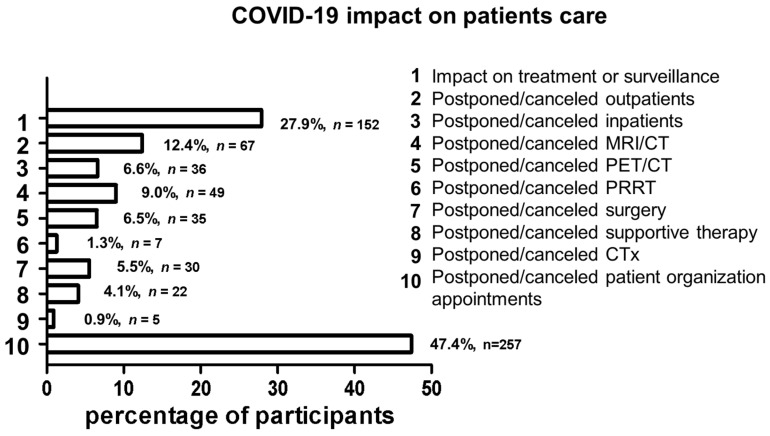
COVID-19 impact on patients care. Absolute and relative patients number are presented for impact on treatment or surveillance (1), postponed/canceled outpatients (2), postponed/canceled inpatients (3), postponed/canceled MRI/CT (4), postponed/canceled PET/CT (5), postponed/canceled PRRT (6), postponed/canceled surgery (7), postponed/canceled supportive therapy (8), postponed/canceled chemotherapy (9) and postponed/canceled patient organization appointments (10).

**Figure 2 cancers-14-00613-f002:**
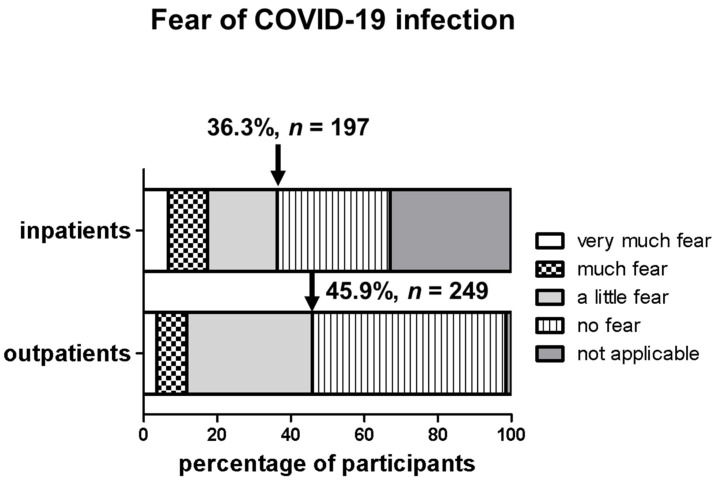
Fear of COVID-19 infection categorized by very much fear, much fear, a little fear, no fear and not applicable for inpatients and outpatients. The black arrow indicates the participants with any kind of fear that were hesitant to visit the NET centers, presented in relative and absolute numbers.

**Figure 3 cancers-14-00613-f003:**
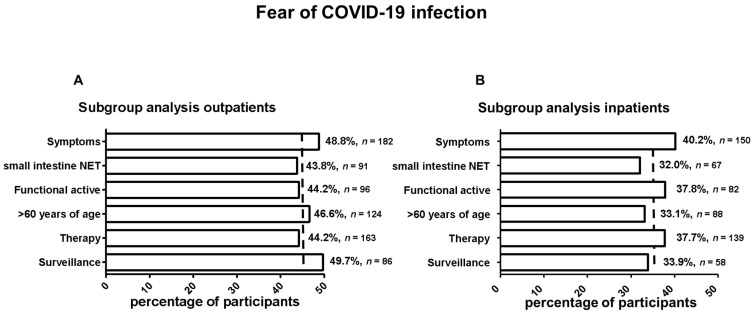
Subgroup analysis for fear of COVID-19 infections for outpatients and inpatients stratified by symptoms, primary tumor localization (small intestine NET = SI-NET), functional activity (FA), age (years of age) and treatment strategy. The dashed line in (**A**,**B**) shows the patient’s fear of COVID-19 infection in the outpatient or inpatient setting.

**Figure 4 cancers-14-00613-f004:**
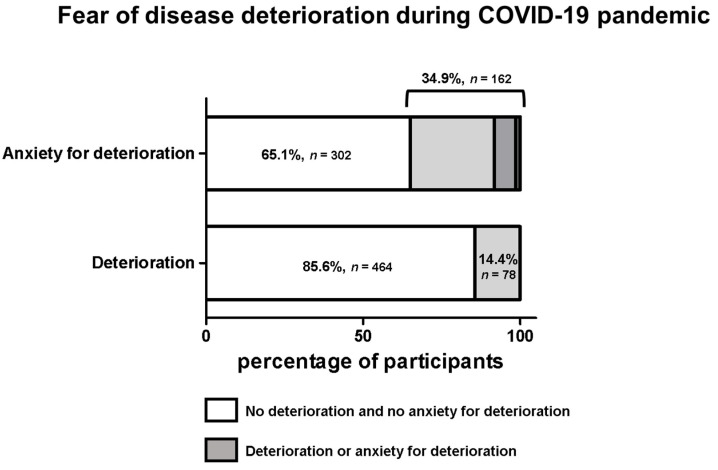
Fear of disease deterioration during the COVID-19 pandemic. The upper panel presents the fear of worsening of NET disease due to the COVID-19 pandemic Represented from left to right: no fear, little fear, much fear and very much fear (shades of grey). The bottom panel shows a subjective worsening of the NET disease due to the COVID-19 pandemic.

**Figure 5 cancers-14-00613-f005:**
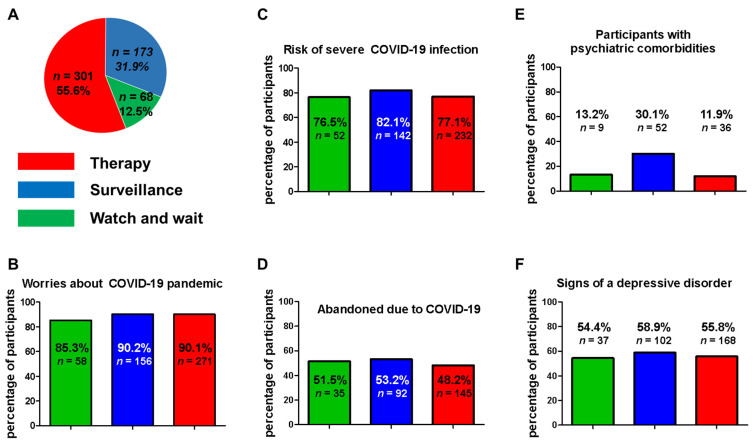
Psychological factors of NET patients during the COVID-19 crisis. Distribution of the NET patients based on their treatment scheme (**A**). Presentation of worries about the COVID-19 pandemic (**B**), risk for severe COVID-19 infection (**C**) and loneliness due to COVID-19 (**D**). Psychiatric comorbidities in NET patients before COVID-19 and signs of a depressive disorder during COVID-19 are shown in (**E**,**F**).

**Figure 6 cancers-14-00613-f006:**
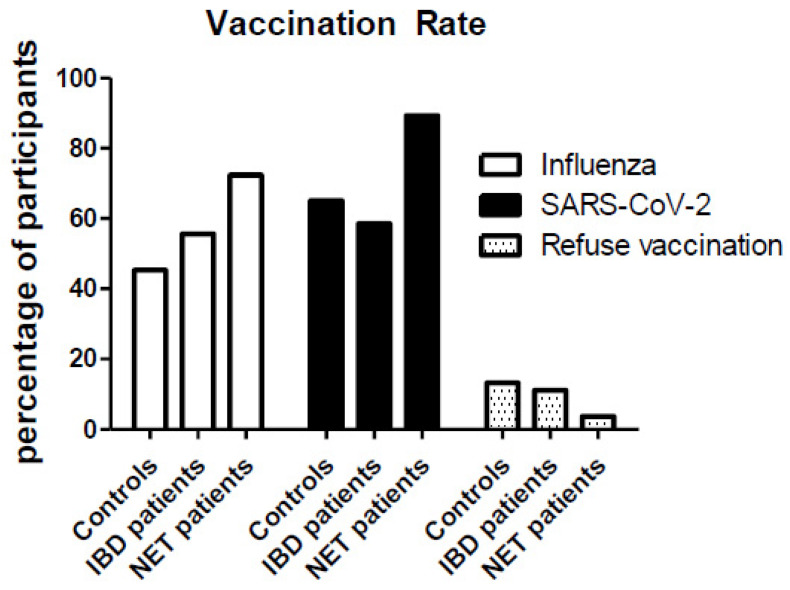
Vaccination rate for influenza (within the last 5 years) and SARS-CoV-2 in a control population, patients with irritable bowel disease and NET patients.

**Table 1 cancers-14-00613-t001:** Clinical characteristics of the participants.

*n* = 542	Therapy *n* = 369	(%) 68.1	Surveillance *n* = 173	(%) 31.9	All	(%)
Age						
18–40	14	3.8	8	4.6	22	4.1
41–60	169	45.8	85	49.1	254	46.9
61–80 and >80	186	50.4	80	46.3	266	49.0
Tumor localization						
small bowel	148	40.1	60	34.7	212	39.1
pancreas	85	23.0	46	26.6	134	24.7
duodenal	36	9.8	14	8.1	50	9.2
lung	23	6.2	15	8.7	41	7.6
CUP	30	8.1	9	5.2	42	7.8
others	47	12.4	29	16.6	66	12.2
Functional active						
yes	159	43.1	58	33.5	217	40.0
no (+unknown)	210	56.9	115	66.5	325	60.0
Symptoms						
yes	269	72.9	104	60.1	373	68.8
impaired resilience	179	66.5	71	68.3	250	67.0
diarrhea	164	60.9	57	54.8	221	59.3
flush	91	33.8	18	17.3	109	29.2
Time of diagnosis						
>5 years	170	46.1	80	46.2	250	46.1
1–5 years	142	38.5	79	45.7	221	40.8
<12 months	57	15.4	14	8.1	71	13.1
Period from symptoms to diagnosis						
<3 months	79	21.4	48	27.8	127	23.4
3–12 months	79	21.4	44	25.4	123	22.7
>12 months	147	39.8	48	27.8	195	36.0
Therapy						
SSA	201	54.5	-	-	201	54.5
PRRT	29	7.9	-	-	29	7.9
CTx	29	7.9	-	-	29	7.9
TKI	15	4.1	-	-	16	4.3
W & W	68	18.4	-	-	68	18.4
Treatment setting						
ENETS center	122	33.1	56	32.4	178	32.8
University Hospital (none ENETS)	136	36.9	71	41.0	207	38.2
Non-university Hospital	49	13.3	15	8.7	64	11.8
Specialist practice	62	16.7	31	17.9	93	17.2

Abbreviations: CUP, cancer of unknown primary; SSA, somatostatin analogues; PRRT, Peptide Receptor Radionuclide Therapy; CTx, chemotherapy; TKI, tyrosine kinase inhibitors; W & W, watch and wait.

**Table 2 cancers-14-00613-t002:** Characteristics of COVID-19 positive patients.

*n* = 20	Therapy *n* = 11	(%) 55.0	Surveillance *n* = 9	(%) 45.0	All	(%)
Age						
18–40	2	18.2	-	-	2	10
41–60	4	36.4	6	66.6	10	50
61–80	5	45.4	3	33.3	8	40
Tumor localization						
small bowel	5	45.4	1	11.1	6	30
pancreas	5	45.4	5	55.6	10	50
others	1	9.1	3	33.3	4	
Functional active						
yes	3	27.3	1	11.1	4	20
no (+unknown)	8	72.7	8	88.9	16	80
Symptoms						
yes	5	45.4	7	78.8	12	60
no	6	55.6	2	22.2	8	40
Therapy						
SSA	5	45.4	-	-	5	25
PRRT	2	18.2	-	-	2	10
CTx	1	9.1	-	-	1	5
TKI	1	9.1	-	-	1	5
W&W	2	18.2	-	-	2	10
Comorbidities						
diabetes	1	9.1	3	33.3	4	20
hypertension	6	55.6	5	55.6	11	55
lung disease	4	36.4	-	-	4	20
chronic renal failure	-	-	1	11.1	1	5
liver cirrhosis	-	-	1	11.1	1	5
chronic infections	1	9.1	1	11.1	2	10
none	2	18.2	1	11.1	3	15
Attributed severe risk for COVID-19 infection						
yes	7	63.6	8	88.9	15	75
no	4	36.4	1	11.1	5	25
COVID-19 disease course						
mild	5	45.4	4	44.4	9	45
moderate	5	45.4	5	55.6	10	50
severe	1	9.1	-	-	1	5
Long-term effects of COVID-19 disease						
fatigue	5	45.4	4	44.4	9	45
impaired concentration	3	27.3	4	44.4	7	35
impaired olfactory/ gustatory sense	1	9.1	3	33.3	4	20
insomnia	-	-	2	22.2	2	10
headaches	2	18.2	-	-	2	10
Vaccination planned						
yes	9	81.8	8	88.9	17	85
undecided	2	18.2	1	11.1	3	15

General questions in white, COVID-19 specific questions in grey, others = 3 × unknown, 1 × lung.

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
