# Peer review of "The Patient’s Point of View: COVID-19 and Neuroendocrine Tumor Disease"

_cancers, 2022, doi:10.3390/cancers14030613_

Round 1

Reviewer 1 Report

MAJOR REVISION

  • From line 223 to 241: this paragraph is a little bit confused; I suggest to write figure five’s concepts better

MINOR REVISIONS

  • From line 121 to 123: I suggest to add these data in table 1
  • From line 126 to 130: considering the importance of these data, I suggest to create a new table or add these data in table 1
  • Figure 1, point 10: I suggest to write “ postponed/canceled patient organization appointments”
  • Line 206: reference to figure 1 is not clear
  • From line 255 to 258: I suggest to add these data in table 2
  • Line 332: I suggest to modify expression “the survey included 20 patients with corona infection”

Reviewer 2 Report

The study of Krug et al. elucidates impacts of the worldwide COVID-19 pandemic on NET-patients, more precisely on their medical care, personal social and psycho-emotional behaviour, and willingness for getting the COVID-19 vaccination. The sample number of n = 542 constitutes a valuable and representative cohort. This study is a very interesting and important approach and a great contribution to improve medical and psychological care of NET patients. Nevertheless, I have a few issues, mainly regarding statistics and figures, as well as some minor comments.

Major points:

  • Material and Methods are quite short compared to the results section. You could think about extending it by briefly describing structure of the questionnaire and statistical methods including p-value etc. Which variables were compared by Chi-square and Fisher’s exact test?
  • Results: Which statistical results/ (significant) differences/ trends between the subgroups did you obtain? (Significant) statistical relations were not explicitly mentioned in the whole results (and discussion) part and overall p-values (if calculated) were not added to the text, figures and tables (except of line 240). It would be beneficial to be more precise here instead of describing too many numbers and percentages and kindly add if relations/trends are statistically significant or not.

Figures:

  • Figure 1: Numbers, letters, and size of the graph in general are quite small; it would be great if you could increase the size.
  • Figure 3: Please improve the labelling of the axis or add a legend: >60 is probably >60 years of age; SI-NET = small intestinal NET. In my opinion, this graph is not appropriate to substantiate the statement made in line 182-183 and it is hard for the reader to get the main point. Which statistical test has been applied for testing trends between subgroups or is this graph exclusively a descriptive one?
  • Same for figure 5, especially headings of graphs summarized as 5B. Please kindly revise/ check all figures, especially axis labelling and size of numbers and letters. Line 221 (Figure 5): Please revise the figure description thoroughly.

Minor points:

  • Line 2-3: You could thing about re-phrasing the title (2x colons) e.g., The Patients’ Point of View: COVID-19 and Neuroendocrine Tumor Disease – High Levels of Anxiety
  • Line 31: Please spell out/ define “SSA”therapy
  • Generally, adapt citation style to the Journal’s guidelines (e.g., [1,2] instead of [1], [2]). Same for figure references.
  • Line 65: Could you specify the term “functionality” once in this context?
  • Line 105: This paragraph is not only about patients’, but also about disease/tumor/clinical characteristics; think about dividing the paragraph or re-wording the heading
  • Line 112: Exchange “the rest” by “one quarter was either divorced, widowed, or single”
  • Line 114-115: Add primary “site” and the “n=” accordingly
  • Line 118: Change the sentence “(…) and in 143 cases (26.4%) functionality was unknown.”
  • Line 120: Probably “somatostatinoma”, not “somatostinoma”?
  • Line 121: “The time of diagnosis was >5 years ago in…” – that would facilitate reading.
  • Line 133: Please kindly add a legend below table 1, spelling out abbreviations, e.g., CUP, SSA
  • Line 134: “3.2 Symptoms and Treatments” – the symptoms have already been presented in paragraph 3.1
  • Line 140: Avoid using abbreviations without spelling them out at least once (e.g., SSA, PRRT…)
  • Line 146: Stick to the past tense when describing your results (“were” instead of “are”)
  • Line 177: Numbers of self-testing are probably given per week?
  • Figure 2: Kindly add to the figure description, that people with any kind of fear were hesitant to visit the NET centers, indicated by the black arrows
  • Figure 4: upper panel: there are four categories, indicated in different grey-shades, but a colour-legend is lacking. Additionally, think about using other colors or pattern, which are easier to distinguish
  • Line 206: The reference to figure 1 is not correct.
  • Line 217: “In 36.9% of cases…” instead of “In 36.9% of the participants…”
  • Table 1 and 2: Add “0” or “-“, if not appropriate.
  • Line 262: Sentence from another figure?
  • Line 272: Delete “respectively”.
  • Line 274: “Remarkably,” or “Notably,” instead of “Of note,” at the beginning of a sentence.
  • Line 276: There is a “n=” missing
  • Line 400: Supplementary Material is not visible to me – does it include the patient’s questionnaire as well or only a survey flow chart as indicated. If not, could you add it?

Round 2

Reviewer 1 Report

MINOR REVISIONS
  • Table 1 and table 2 are completed now, but I suggest to replace them with sharper images.
  • In discussion paragraph, I suggest to cite Italian work “The psychological impact of COVID-19 pandemic on patients with neuroendocrine tumors: Between resilience and vulnerability”. This may be an additional discussion element on Covid-19 impact in NET patients. 

Reviewer 2 Report

Thank you very much for the revised version. I still have some problems in understanding table 2 (I think there is a color coding in the annotation, which is not reflected in the table as it appears in the journal documents? I think general questions with white background and COVID 19 specific in grey?)
